# Elasticity of podosome actin networks produces nanonewton protrusive forces

Marion Jasnin [1✉], Jordan Hervy[2], Stéphanie Balor[3], Anaïs Bouissou[4], Amsha Proag [4], Raphaël Voituriez [5], Jonathan Schneider[1], Thomas Mangeat[6], Isabelle Maridonneau-Parini[4], Wolfgang Baumeister [1], Serge Dmitrieff [2✉] & Renaud Poincloux [4✉]

Actin filaments assemble into force-generating systems involved in diverse cellular functions, including cell motility, adhesion, contractility and division. It remains unclear how networks of actin filaments, which individually generate piconewton forces, can produce forces reaching tens of nanonewtons. Here we use in situ cryo-electron tomography to unveil how the nanoscale architecture of macrophage podosomes enables basal membrane protrusion. We show that the sum of the actin polymerization forces at the membrane is not sufficient to explain podosome protrusive forces. Quantitative analysis of podosome organization demonstrates that the core is composed of a dense network of bent actin filaments storing elastic energy. Theoretical modelling of the network as a spring-loaded elastic material reveals that it exerts forces of a few tens of nanonewtons, in a range similar to that evaluated experimentally. Thus, taking into account not only the interface with the membrane but also the bulk of the network, is crucial to understand force generation by actin machineries. Our integrative approach sheds light on the elastic behavior of dense actin networks and opens new avenues to understand force production inside cells.

[1] Department of Molecular Structural Biology, Max Planck Institute of Biochemistry, Martinsried, Germany. [2] Université de Paris, CNRS, Institut Jacques Monod, Paris, France. [3] Plateforme de Microscopie Électronique Intégrative, Centre de Biologie Intégrative, CNRS, UPS, Toulouse, France. [4] Institut de Pharmacologie et de Biologie Structurale, Université de Toulouse, CNRS, UPS, Toulouse, France. [5] Laboratoire Jean Perrin, CNRS, Sorbonne Université, Paris, France. [6] LITC Core Facility, Centre de Biologie Integrative, Université de Toulouse, CNRS, UPS, 31062 Toulouse, France. ✉email: jasnin@biochem.mpg.de; serge.dmitrieff@ijm.fr; renaud.poincloux@ipbs.fr

Actin, one of the most abundant proteins in eukaryotic cells, organize into force-generating filamentous networks, which play pivotal roles in cell motility, adhesion, endocytosis, and vesicular traffic[1,2]. Thermodynamics showed that actin polymerization generates mechanical forces by the addition of new monomers at the end of a fluctuating filament[3,4]. Typical stall forces of a polymerizing actin filament were estimated in the 1–10 pN range[3,5,6], in agreement with experimental values of 1.5 pN found by optical trap measurements[7]. Thus, polymerization of actin filaments against a membrane is capable of extruding thin membrane tubes or forming plasma membrane invaginations in mammalian cells in a force range of a few tens of pN[8–10].

The forces exerted by actin filaments can also reach a much higher regime, in the nanonewton range[11–13]. At the leading edge of motile cells, branched actin networks in lamellipodial protrusions produce local forces of ~1 nN[14,15]. During yeast endocytosis, the actin machinery generates forces in a similar range to overcome the turgor pressure pushing the invagination outwards[16]. At the basal membrane of myeloid cells, podosomes probe the stiffness of the extracellular environment through the generation of forces reaching tens of nNs[17]. Unlike the lower force regime, the mechanisms by which meshworks of actin filaments produce nN forces remain unknown.

If the force generated by an actin network corresponds to the sum of the polymerization forces generated by single filaments pushing against a load, then hundreds to thousands of filaments would need to continuously grow against the surface to reach the nN range. Alternatively, filaments could push on the membrane with shallow angles[18,19]. It was also proposed that the actin network could store elastic energy, and thus exert a restoring force. In the low-force regime (~10–100 pN), bending of endocytic filaments observed in animal cells has been proposed to store elastic energy for pit internalization[20]. In the large force regime (~1–10 nN), a theoretical model with non-deformable filaments showed that a large force would cause elastic energy to be stored in cross-linkers deformation[21]. However, none of these hypotheses have been explored in native actin machineries generating forces of several nNs, due to the difficulty to combine direct observation of the actin network architecture and knowledge of the exerted force.

To date, cryo-electron tomography (cryo-ET) is the only technique that resolves single actin filaments inside unperturbed cellular environments[22–24]. Here, we use cryo-ET to unveil the three-dimensional (3D) architecture of human macrophage podosomes and elucidate their force generation mechanism. These submicrometric structures are composed of a protrusive core of actin filaments surrounded by an adhesion ring. The balance of forces requires the protrusion force applied by the core on the substrate to be counteracted by a force of equal magnitude. Protrusion force microscopy (PFM) revealed that this balance of forces takes place locally through traction at the adhesion ring[17,25,26], which has been proposed to be transmitted by radial actin cables connecting the core to the ring[26]. Here, we visualize these radial filaments and quantitatively analyze the 3D organization of the core and ring networks. We show that the protrusive forces generated by podosomes cannot be explained by the sum of the actin polymerization forces at the core membrane but by the storage of elastic energy into the dense network of bent actin filaments. These results explain how cellular actin networks can act as a spring-loaded elastic material to exert forces of up to tens of nNs.

## Results

**Cryo-ET allows quantitative analysis of filament organization in podosomes.** Owing to their size and location at the basal cell membrane (Supplementary Movies 1 and 2), native podosomes are amenable to cryo-ET exploration using cryo-focused ion beam (cryo-FIB) milling sample preparation. We prepared thin vitrified sections (so-called wedges) containing podosomes using shallow incidence angles of the ion beam and subjected them to cryo-ET (Fig. 1a and Supplementary Movie 3). Segmentation of the tomograms revealed that podosomes are made of a core of oblique filaments surrounded by radial filaments (Fig. 1b–d and Supplementary Movie 3), as predicted previously[26]. Other cytoskeletal elements, cellular organelles, ribosomes and glycogen granules are excluded from the core and radial actin networks, gathering either at the periphery or on top of podosomes (Fig. 1a, b and Supplementary Movie 3).

Quantitative analysis of filament organization highlighted the specificity of the actin core in terms of filament length, density and orientation with respect to the basal membrane (Methods and Supplementary Figs. 1–3). All of these parameters exhibit a sharp transition as a function of the radial distance from the core center (Fig. 1e–g). Density is higher inside the core by a factor of 2 to 3 (Fig. 1e) and filaments display a mean orientation of $47 \pm 22°$ relative to the plasma membrane, as compared to the flatter $23 \pm 21°$ outside the core (Fig. 1f and Supplementary Fig. 4a–d). Core filaments are shorter than the surrounding radial filaments, with mean lengths of $111 \pm 46$ nm and $166 \pm 120$ nm, respectively (Fig. 1g and Supplementary Fig. 4e–h). Further analysis of the transition curves for the filament orientation provided a mean core radius of $203 \pm 38$ nm for a total of ten podosomes (Fig. 1f, inset). This agrees with the values obtained from the fits of the other parameters (Supplementary Fig. 5).

Since the milling procedure removed the top of the podosomes, we also imaged podosomes exposed by cell unroofing prior to cryo-fixation to get a complete picture of podosome organization (Supplementary Fig. 6 and Supplementary Movie 4). The distribution of filament length, orientation and density are similar between unroofed and in situ podosomes (Supplementary Fig. 7), indicating that podosome architecture is well preserved after the unroofing procedure. In addition to the core and radial filaments observed previously, we detected horizontal filaments on top of the core as well as in between neighboring cores. These filaments, which may have been lifted up from the plasma membrane during podosome growth, could also participate in the generation of the traction forces that counterbalance the protrusion forces generated by podosomes[26].

**The sum of the actin polymerization forces at the core membrane is below 1 nN.** Actin polymerization can exert forces in the range of 1–10 pN in the direction of polymerization[18]. The effective force against a load depends on the orientation of the filament relative to the load: the shallower the angle of a filament, the higher the effective force[18]. Thus, to generate a force on the order of 10 nN, several thousands of filaments, or filaments with very shallow angles, must be involved[19]. To evaluate the forces generated by actin polymerization on the plasma membrane beneath the podosome core, where protrusion occurs[17,25,26], we identified filament segments in the close vicinity of and protruding against the plasma membrane (Fig. 2a, b and Supplementary Movie 5). We found an average of $45 \pm 29$ filaments per podosome, with a mean orientation of $61 \pm 6°$ relative to the plasma membrane (Fig. 2c). Using the upper limit of 10 pN for the stall force of a single filament growing perpendicularly to the membrane[3,5,6,27], taking into account filament orientation[18], and considering that all these filaments are polymerizing concomitantly, we evaluated a maximal polymerization force of $564 \pm 305$ pN per podosome (ten podosomes evaluated; Methods and Fig. 2d, e). This force would be 56 pN on average if the

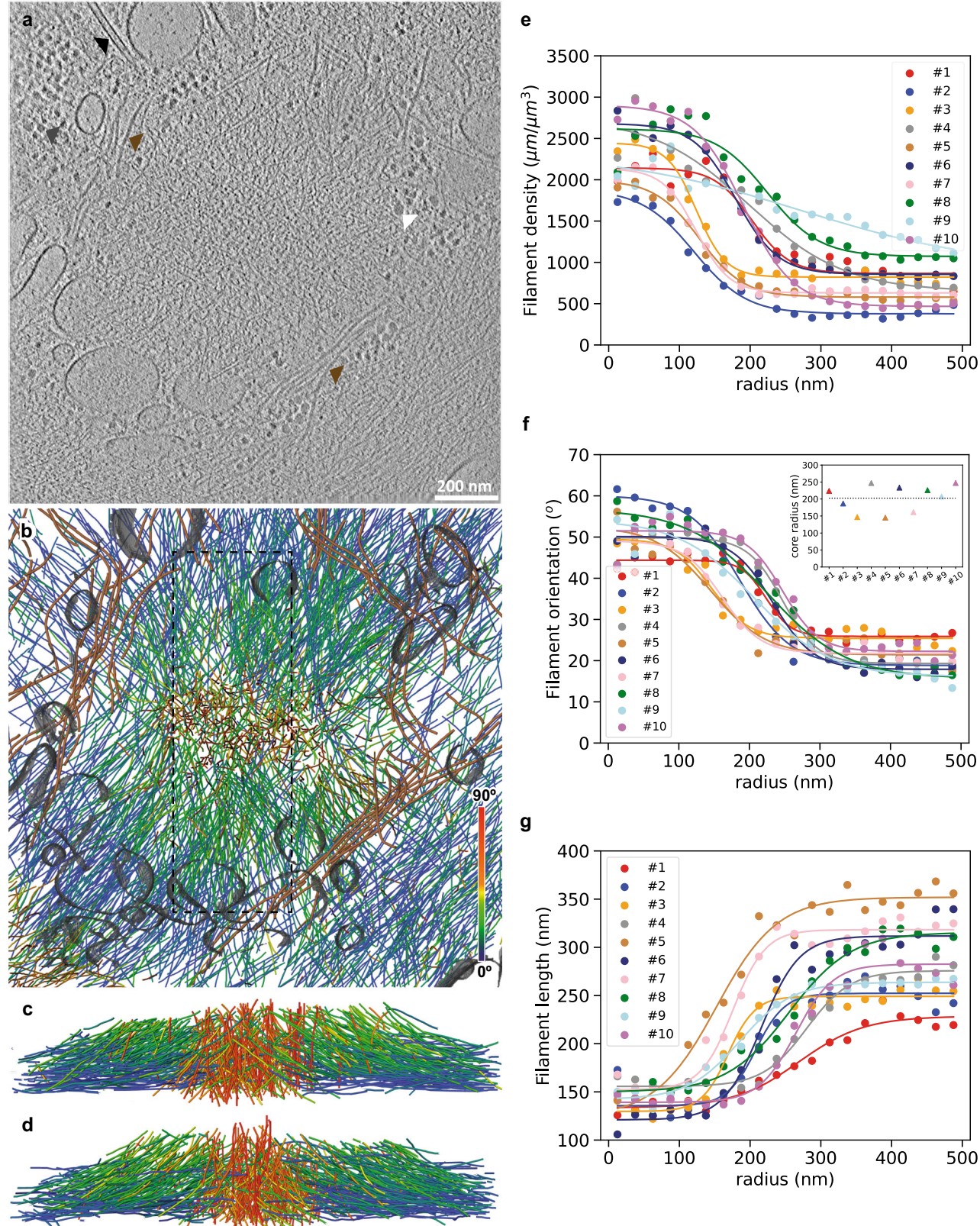

polymerization force was 1 pN, a common lower bound. This is one to two orders of magnitude lower than the experimental values[17]. We note that, because this result depends only on the polymerization force and the observed filament angle with the membrane[18], whether the actin nucleation pattern is branched or linear has no influence. In addition, small filaments in vicinity of

the plasma membrane, which may have been removed to avoid false positives, could contribute to increasing the polymerization force but would not change its order of magnitude. We therefore concluded that the total force produced by polymerization of actin filaments against the plasma membrane is not sufficient to generate the protrusive forces exerted by podosomes.

**Fig. 1 Cryo-electron tomography allows quantitative analysis of actin filament organization in native podosomes. a** Slice from a tomographic volume acquired in a frozen-hydrated human macrophage revealing the podosome environment. Colored arrows point to ribosomes (gray), glycogen granules (white), a microtubule (black), and intermediate filaments (brown). See also Supplementary Movie 3. The original tomogram has been deposited in the EMDB under accession code EMD-13671. Nine other tomograms showing similar organizations were obtained. **b** Orthographic view of the corresponding 3D segmentation of the actin filaments showing their relative orientation with respect to the basal membrane, intermediate filaments (brown), and organelle membranes (gray). **c, d** Perspective views of the actin filaments in the volume indicated by a dotted rectangle in **b** shown from the left (**c**) and right (**d**) sides. **e–g** Average filament density (**e**), orientation (**f**), and length (**g**) as a function of the radial distance from the core center for ten tomograms. Inset in **f**: Corresponding values estimated for the core radius (Methods). The podosome shown in **a–d** corresponds to #9. Source data are provided as Source Data files.

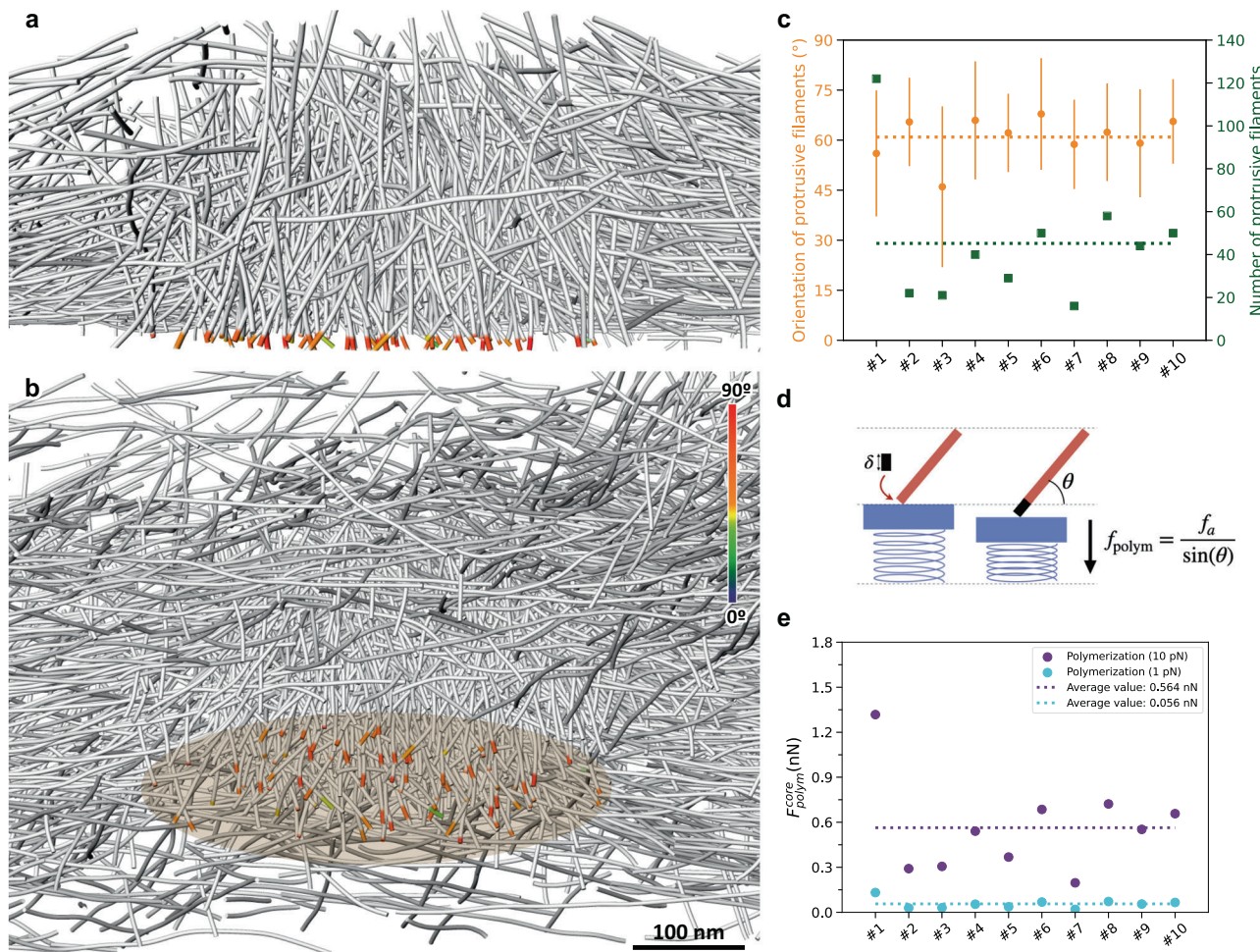

**Fig. 2 The sum of the actin polymerization forces at the core membrane is below 1 nN. a, b** Perspective view (**a**) of a cross-section through podosome #8 displaying in color the part of the core filaments pushing against the plasma membrane, and orthographic view (**b**) of the same podosome from a different angle. The color map corresponds to the filament segment orientation with respect to the basal membrane. See also Supplementary Movie 5. The core diameter is approximately 400 nm. The original tomogram has been deposited in the EMDB under accession code EMD-13669. **c** Mean orientation (orange) and number (green) of polymerizing filaments at the core membrane. Data are presented as mean values +/− SD. At least 16 filaments were measured by tomogram. **d** Scheme representing the polymerization force $f_{polym}$ generated by the addition of a new monomer at the growing end of a filament with an inclination $\theta$ relative to the membrane. **e** Estimated polymerization force generated at the core membrane ("Polymerization") using the lower ("1 pN"; cyan filled circles) and upper ("10 pN"; purple filled circles) limit of the stall force, and respective average values (dotted lines of the same color). Source data are provided as Source Data files.

**The actin core stores elastic energy**. Next, we tested the third hypothesis, that is, the storage of elastic energy in the network. Visual inspection of the networks indicated that podosome filaments are bent (Fig. 3a and Supplementary Movie 6). The projection of each filament on the vertical plane passing through its two ends highlighted the variation of their local curvature along the filament length (Methods and Fig. 3b–d). For each filament in the podosomes, we computed the orientational correlation length, which is expected to be on the order of 10 μm for thermally fluctuating filaments[28]. We find a correlation length of 1.68 μm on average in the core, and 2.41 μm outside the core, showing that actin filaments are actively deformed in podosomes, especially in the core (Fig. 3e).

We therefore evaluated the elastic energy using the theory of linear elasticity: actin filaments can be modeled as semi-flexible polymers with the following elastic energy at the single filament

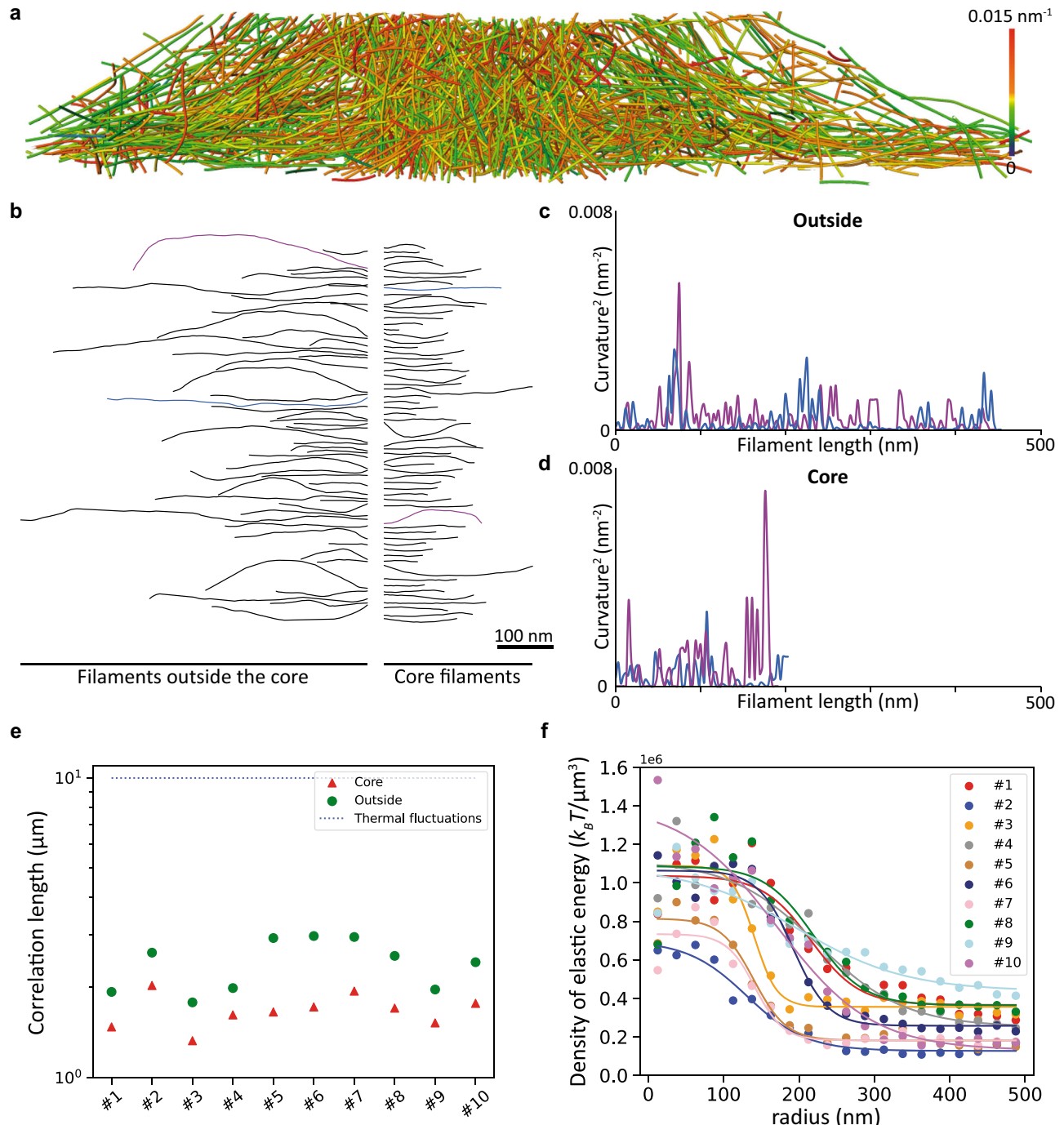

**Fig. 3 Podosome filaments are compressed and store high elastic energy inside the core. a** Perspective view of a cross-section through podosome #1 displaying the mean curvature of the filaments. See also Supplementary Movie 6. The network diameter is 1.56 µm. The original tomogram has been deposited in the EMDB under accession code EMD-13666. **b** Gallery of a random selection of actin filaments from podosome #1. Filaments from the core (right) and outside the core (left) are shown. **c**, **d** Square of the local curvature along the filament length for the blue and purple filaments from the galleries of filaments outside (**c**) and inside (**d**) the core shown in **b**. **e** Correlation length in the ten observed podosomes. The correlation length inside the podosome core (1.68 µm on average) is smaller than outside (2.41 µm on average). Both values are smaller than those expected from thermal fluctuations of the filaments (about 10 µm[30]), showing that energy has been spent to compress the filaments. **f** Density of elastic energy as a function of the radial distance from the core center for ten tomograms. Source data are provided as Source Data files.

level[29]:

$$u_{\text{elastic}} = \frac{\kappa}{2} \int \left\| \frac{\partial t(s)}{\partial s} \right\|^2 ds \qquad (1)$$

where $\kappa \sim 4*10^{-26}$ N.m² is the bending modulus[30], $t(s)$ is the tangent vector as a function of the arc-length coordinate $s$, and the

integrand represents the square of the local curvature along the filament (Methods and Supplementary Fig. 8a). We evaluated the elastic energy per unit volume as the sum of the energies over all filaments in a given interval at a radial distance $r$ from the core (Supplementary Fig. 8b). The density of elastic energy stored inside the core is much larger than that outside the core, by factors

ranging from 3 to 10 (Fig. 3f). This is consistent with the larger filament density by factors of 2 to 6 measured inside the core (Fig. 1e). Therefore, this specific architecture allows the system to store elastic energy inside the podosome core, ranging from ~$10^4$ $k_BT$ to ~$5.10^4$ $k_BT$, and corresponding to $40k_BT$ per filament on average. In addition, we expect elastic energy to be stored in cross-linkers, as they are expected to be stretched and compressed by network deformations[21].

**The actin core generates an elastic force in the nN range.** To test whether the stored elastic energy can account for podosome protrusion forces, we next evaluated the elastic force generated through the compression of the actin network. First, we computed the compressive strain $\varepsilon_{core}$ of the podosome cores, a dimensionless number between 0 (undeformed network) and 1 (fully compressed network). As can be done for an assembly of springs in series or in parallel, we assumed $\varepsilon_{core}$ to be the average filament strain, which we could compute from the data (Supplementary Fig. 9), and found $\varepsilon_{core} = 0.016 \pm 0.02$, i.e., a 1.6% deformation.

Following previous work, we then assumed that the podosome core behaves as a homogeneous elastic material[31,32]. As the average compressive strain is small, we can also assume elasticity to be linear, and that the force exerted by the core is such that its work for a small deformation of amplitude $\delta h_{core}$ is equal to $U_{elastic}^{core}/\delta h_{core}$ (with $U_{elastic}^{core}$ the total elastic energy stored in the podosome core). With $\delta h_{core} = \varepsilon_{core} \times h_{core}$ ($h_{core}$ being the core height, see Methods), we find:

$$F_{elastic}^{core} \approx \frac{U_{elastic}^{core}}{\varepsilon_{core} \times h_{core}} \qquad (2)$$

Note that because of internal constraints such as entanglements and crosslinking, not all of the elastic energy stored in the core can be released when the stress is relaxed, and this force value is likely to be overestimated. Using Eq. 2, we found an average elastic force of $27.7 \pm 11.8$ nN, higher but of the same order as the mean value of $10.4 \pm 3.8$ nN reported from PFM measurements on Formvar[17]. We note that Formvar is softer than the carbon films of the EM grids used for cryo-ET exploration; as podosomes are mechanosensitive[25], the forces on the carbon films are expected to be higher. The elastic force varies significantly between podosomes, with values ranging from 11.5 to 45.2 nN (Fig. 4a). The force per unit area of the core exhibits much less variation with an average value of $P = 202.4 \pm 29.5$ kPa (Methods and Fig. 4b), which suggests that the force of a podosome is regulated primarily by its size. This is higher but of the same order of magnitude as the pressure estimated by PFM on Formvar[17] (Methods and Fig. 4b) and five times higher than the pressure estimated during endocytosis in yeast[19]. We mentioned earlier that the estimated force was likely to be overestimated because not all the bending energy can be released. On the other hand, we do not take into account the elastic energy stored by cross-linkers[21] and Arp2/3 complexes. Recent work has shown that storage of elastic energy by cross-linkers could result in a compressive force[21]. This would contribute to increase the stored elastic energy and thus the elastic force. Small filaments discarded by the segmentation method could also marginally contribute to increasing the elastic energy and thus the force. Therefore, this computation provides an order-of-magnitude estimate of the force, which is in agreement with the PFM measurements.

**Mechanical properties of the podosome.** Knowing the elastic force, and thus the pressure $P$, allowed us to estimate the Young's modulus of the core: $Y = P/\varepsilon_{core} = 13.2 \pm 2.3$ MPa. While this value is several orders of magnitude larger than reported values for reconstituted and cellular actin networks[32,33], it is compatible

with the very high actin density in the core. Actin itself has a Young's modulus $Y_a = 2.3$ GPa[28]. The elastic modulus that can be reached by an actin network can be estimated as $Y = Y_a\phi^2$, with $\phi$ the volume fraction occupied by actin filaments[34], yielding a value of 12 MPa in podosomes. The abundance of actin-binding proteins such as cross-linkers and Arp2/3 complexes would effectively increase $\phi$. Thus, the Young's modulus we find for the podosome core is well within expected values and helps understand how a compressive strain of < 2% translates into forces in the nanonewton range.

The protrusive force exerted by the core on the substrate is balanced by an opposing force transmitted by the radial filaments to the adhesion ring, which prevents the core from relaxing towards the cell interior. We can estimate the surface tension, $\sigma$, of the 2D meshwork of radial filaments that is required to balance the force $F_{elastic}^{core}$, knowing the mean radius of the core, $r_{core}$, and the mean angle of the radial filaments, $\theta_{radial}$, with respect to the membrane plane (Fig. 4c and Supplementary Fig. 4c, d):

$$\sigma = \frac{F_{elastic}^{core}}{2\pi r_{core}\sin\theta_{radial}} \sim 56 \text{ mN/m} \qquad (3)$$

While this tension is one order of magnitude larger than the cortical tension of rounding mitotic cells[35], it is the same order of magnitude as the tension inferred for some adhering cells, from 6 mN/m[36] to 100 mN/m[37].

Finally, we sought to perturb the system and see if our force model agrees with the experimental force measurements. Since our force estimates scale with actin density, we decreased actin density experimentally using cytochalasin D, which we have previously shown to almost completely abolish protrusion force generation[25], unroofed and cryo-fixed the cells. We showed that podosome architecture is well preserved after the unroofing procedure (Supplementary Fig. 7); we note, however, that unroofed podosomes have higher correlation lengths in the core as compared to in situ podosomes (2.30 μm versus 1.67 μm; Fig. 3e and Supplementary Fig. 10), indicating that filaments are less compressed after unroofing than in their native context, and that the unroofing procedure may have released some of the constraints in the core. Our cryo-ET observations showed that the network of radial filaments disappeared after cytochalasin D treatment (Supplementary Fig. 11), confirming previous observations by fluorescence microscopy[25]. Because of force balance, this lack of radial filaments should strongly decrease the protrusion force in cytochalasin D-treated cells, as was measured experimentally[25]. Accordingly, the remaining cores are composed of less dense filament networks, the correlation lengths are larger (2.58 μm in the core) and the compressive strain decreased by 50% as compared to the control unroofed cells (Supplementary Fig. 12). This indicates that the release of radial tension leads to the release of filament compression, in agreement with our model.

## Discussion

In summary, our results showed that mechanical energy accumulates inside the podosome core through bending of the dense network of actin filaments (Figs. 1 and 3). To relax, the elastic energy produces an elastic force, which is balanced by radial tension between the actin core and the adhesion ring. Indeed, we found that the elastic force computed from the bending energy is of the same order of magnitude as the protrusive force measured experimentally (Fig. 4).

Our results imply that the core filaments do not act independently but are mechanically linked to each other. The Arp2/3 complex, as well as fascin and L-plastin, are present in the core[38,39] and are likely to participate in network connectivity. α-actinin and filamin-A presumably connect the radial filaments to each other,

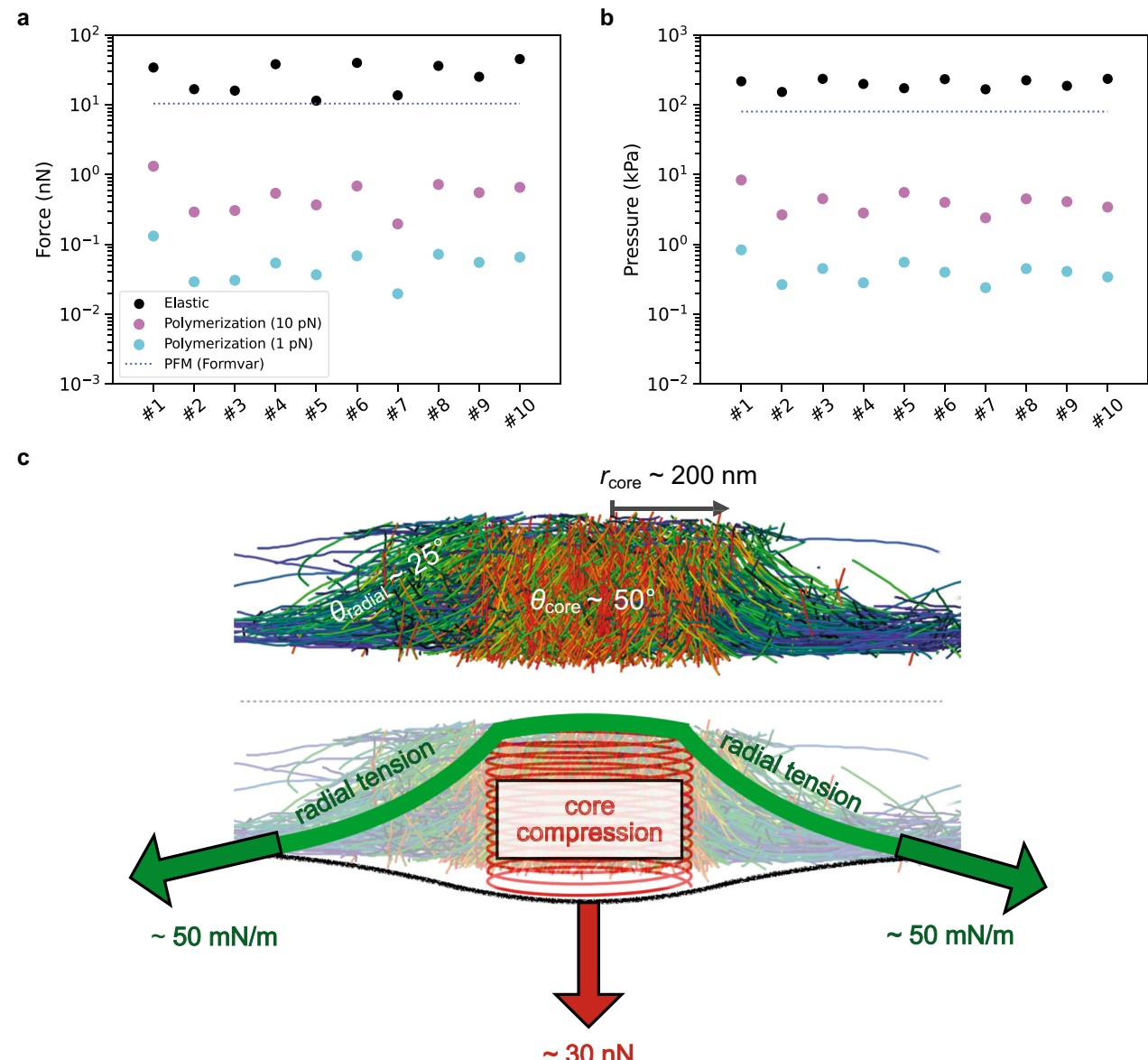

**Fig. 4 The actin core generates elastic forces in the nanonewton range. a** Comparison of the elastic force generated by the core through the compression of the actin network ("Elastic"; black) with the estimated polymerization force generated at the core membrane ("Polymerization") with the upper ("10 pN"; purple) and lower ("1 pN"; cyan) limits of the polymerization force. The blue dashed line corresponds to the mean force derived from PFM measurements on 30-nm thick Formvar films ("PFM (Formvar)")[17]. **b** Same comparison plot as in **a** for the estimated pressure assuming a perfect circular shape for the podosome core (Methods). **c** Summary scheme showing the podosome organization revealed by cryo-ET, the elastic force resulting from core compression and the radial surface tension counterbalancing it. The tension of the 2D meshwork of radial filaments that is required to balance the core compression force is deduced from the mean radius of the core, $r_{core}$, and the mean angle of the radial filaments, $\theta_{radial}$, with respect to the membrane plane. Source data are provided as Source Data files.

but also possibly to the core filaments[40,41]. Finally, actin filaments have also been proposed to be connected to the substrate at the adhesion ring via integrins, talin and vinculin[26]. All of these actin interactors therefore represent mechanical connections that are also subjected to forces and can also store elastic energy.

This work also suggests that the load borne by actin filaments in contact with the membrane is greater than the stall force, making their growth thermodynamically unfavorable (Fig. 2). In addition, the need to synchronize their growth against the membrane would drastically slow down the polymerization process[42]. Thus, how the network assembles under force remains to be understood.

One possibility is that actin filaments do not grow directly against the load at the membrane, which would instead be borne by the existing dense and compressed network. They would rather grow bent in the bulk of the podosome core in a dense environment, thus increasing the elastic energy stored by the network, which can then relax by pushing against the membrane. An estimation of the energy released by actin polymerization in the core, by summing over all actin monomers[18], yields $1 - 3.10^5 k_B T$, which is one order of magnitude larger than the elastic energy $U_{elastic}^{core}$. While some of the polymerization energy will be dissipated as heat, the polymerization reaction can still significantly contribute to the loading of the network.

Another possibility is that the filaments grow under a much smaller force, and that the network is then loaded by the active tension of the actomyosin cables driven by myosin II. This would be consistent with the decrease in pushing force when myosin II is inhibited[25]. In this scenario, the energy source would be mostly ATP consumed by myosin II rather than the polymerization energy. We expect that a combination of experiments and simulation will be able to clarify the assembly of such a spring-loaded network.

In conclusion, to understand the forces applied by actin assemblies, it is not sufficient to consider only the local actin-membrane interactions; rather, the whole system must be taken into account. Through a combination of quantitative analysis and modeling of the native 3D architecture of macrophage podosomes, we showed that elastic energy is stored in actin networks in vivo, allowing forces of the order of tens of nN to be produced. These results highlight the possibilities opened by exploring the architecture of actin networks in situ at the molecular scale, particularly those generating large forces on their environment, such as branched actin networks in lamellipodial protrusions or those allowing yeast endocytosis. Given the rapid progress of cryo-ET, we expect that, building on our biophysical approach, it will soon be possible to address other degrees of freedom for elastic energy storage, such as crosslinker deformation and filament twisting, and to shed light on other modes of force generation by cellular machineries.

## Methods

### Differentiation and culture of primary monocyte-derived macrophages.
Human peripheral blood mononuclear cells were isolated from the blood of healthy donors by centrifugation through Ficoll-Paque Plus (Dutscher), resuspended in cold phosphate buffered saline (PBS) supplemented with 2 mM EDTA, 0.5% heat-inactivated Fetal Calf Serum (FCS) at pH 7.4 and monocytes were magnetically sorted with magnetic microbeads coupled with antibodies directed against CD14 (Miltenyi Biotec # 130-050-201). Monocytes were then seeded on glass coverslips at $1.5 \times 10^6$ cells/well in six-well plates in RPMI 1640 (Invitrogen) without FCS. After 2 h at 37 °C in a humidified 5% $CO_2$ atmosphere, the medium was replaced by RPMI containing 10% FCS and 20 ng/mL of Macrophage Colony-Stimulating Factor (M-CSF) (Peprotech). For experiments, cells were harvested at day 7 using trypsin-EDTA (Fisher Scientific) and centrifugation (320 g, 10 min).

### Random illumination microscopy.
For live imaging shown in Supplementary Movies 1, 2, macrophages were detached using trypsin-EDTA and transfected with Lifeact-GFP +/− Zyxin-mCherry plasmids 4 h before observation using a Neon® MP5000 electroporation system (Invitrogen) with the following parameters: two 1000 V, 40 ms pulses, with 0.5 μg DNA each for $2 \cdot 10^5$ cells.

Random illumination microscopy was then performed using a home-made system described in ref. [43]. Briefly, images were acquired every 12 ms using an inverted microscope (TEi Nikon) equipped with a 100x magnification, 1.49 N.A. objective (CFI SR APO 100XH ON 1.49 NIKON) and an sCMOS camera (ORCA-Flash4.0 LT, Hamamatsu). Fast diode lasers (Oxxius) with respective wavelengths 488 nm (LBX-488-200-CSB) and 561 nm (LMX-561L-200-COL) were collimated using a fiber collimator (RGBV Fiber Collimators 60FC Sukhamburg) to produce TEM00 2.2 mm-diameter beam. The polarization beam was rotated with an angle of 5 degrees before hitting a X4 Beam Expander beam (GBE04-A) and produced a 8.8 mm TEM00 beam. A fast spatial light phase binary modulator (QXGA fourth dimensions) was conjugated to the image plane to create speckle random illumination. Image reconstruction was then performed as detailed in ref. [43] and at https://github.com/teamRIM/tutoRIM.

### Cell vitrification.
Gold EM grids with Quantifoil R 1.2/20 or R 1/4 holey carbon film (or R 1/4 holey $S_iO_2$ film for the cytochalasin D data; Quantifoil Micro Tools GmbH) were glow-discharged in an EasiGlow (Pelco) glow discharge system. After grid sterilization under UV light, the cell suspension containing fiducials was seeded onto the grids and incubated for 2 h at 37 °C to let the cells adhere to the grids, resulting in 3 to 4 cells per grid square. For cell vitrification, grids were loaded into the thermostatic chamber of a Leica EM-GP automatic plunge freezer, set at 20 °C and 95% humidity. Excess solution was blotted away for 10 s with a Whatman filter paper no. 1, and the grids were immediately flash frozen in liquid ethane cooled at −185 °C.

### Unroofing and cytochalasin D treatment.
When indicated, macrophages plated on grids were unroofed prior to vitrification. Cells were unroofed using distilled water containing cOmplete™ protease inhibitors (Roche) and 10 μg/mL phalloidin (Sigma-Aldrich P2141) for 30 s.

For Supplementary Figs. 11 and 12, cells were treated with 2 μM cytochalasin D (Sigma) for 10 min before unroofing.

### Cryo-FIB milling.
Plunge-frozen EM grids were clipped into Autogrid frames modified for wedge milling under shallow angles[44]. Autogrids were mounted into a custom-built FIB-shuttle and transferred using a cryo-transfer system (PP3000T, Quorum) to the cryo-stage of a dual-beam Quanta 3D FIB/SEM (Thermo Fisher Scientific) operated at liquid nitrogen temperature[45]. The support film close to the cells of interest was sputtered away with high-beam currents of 0.5–1.0 nA to provide a reference in Z direction for wedge milling. Cells were first milled roughly at very shallow angles (typically 2–5° of the incident ion beam) with beam currents of 300–500 pA. Advancement of the milling was monitored by SEM at 5 kV and 5.92 pA. Closer to the cell surface, beam currents of 50–100 pA were used for fine milling. Once all the wedges were prepared on the grid, a final polishing step at 30–50 pA was performed to limit surface contamination.

### Cryo-ET and tomogram reconstruction.
Wedges were loaded vertically to the tilt axis in a Titan Krios transmission electron microscope (Thermo Fisher Scientific) equipped with a 300 kV field-emission gun, Volta phase plates (VPPs)[46], a post-column energy filter (Gatan, Pleasanton, CA, USA) and a 4k × 4k K2 Summit direct electron detector (Gatan) operated with SerialEM. The VPP was on-plane and the beam-shift pivot points aligned on the phase-plate plane[47].

*In situ data and unroofed data without treatment.* Low-magnification images were recorded at 2250x; high-magnification tilt-series were recorded in counting mode at 33,000x (calibrated pixel size 0.421 nm) with a target defocus for phase-plate imaging of 0 or −0.1 μm (that is, nominal values of −0.270 and −0.370 μm, respectively). Bi-directional tilt-series were acquired typically from −30° to +60° and −32° to −60° with a tilt increment of 2° and a total dose between 150 and 200 e-/Å². Frames were aligned with in-house software K2Align (https://github.com/dtegunov/k2align) based on procedures developed by Li et al.[48].

*Unroofed data with cytochalasin D treatment.* Low-magnification images were recorded at 6500x; high-magnification tilt-series were recorded in counting mode at 42,000x (calibrated pixel size 0.342 nm) with a target defocus range of −4 to −5 μm, from −50° to +50° with 2° steps, and a total dose of 200 e−/Å². Pre-processing was performed using the TOMOMAN package (https://github.com/williamnwan/TOMOMAN) as follows: frames were aligned using MotionCor2[49] and dose-filtered by cumulative dose using the exposure-dependent attenuation function and critical exposure constants described in ref. [50] and adapted for tilt-series in refs. [51,52]. Tilt-series were aligned using the gold beads deposited on the surface of the support film as fiducial markers. 3D reconstructions with final pixel sizes of 1.684 nm and 1.368 nm, respectively, were obtained by weighted-back projection using the IMOD software[53]. To enhance contrast, the cytochalasin D tomograms were denoised with cryo-CARE[54]. A total of ten in situ tomograms (collected from five different cells), four tomograms of podosomes in untreated unroofed cells (collected from two different cells), and four tomograms of podosomes in unroofed cells treated with cytochalasin D (collected from one cell) were used in this study.

### Automated filament segmentation.
The 4× binned tomograms (pixel size of 1.684 nm or 1.368 nm) were subjected to non-local-means filtering using the Amira software[55] provided by Thermo Fisher Scientific. Actin filaments were traced using an automated segmentation algorithm based on a generic filament as a template[56], with a diameter of 8 nm and a length of 42 nm. To reduce background noise, short filamentous structures with lengths below 60 nm (or 50 nm for the unroofed podosomes) were filtered out.

The coordinates of the segmented filaments were exported from Amira and used as input for data analysis in MATLAB (The MathWorks) and in python (Podosome analyzer, https://gitlab.com/SergeDmi/podosome-demo[57]). The coordinates of the filaments were resampled every 3 nm to give the same weight to every point along a filament[22].

### Radial-distance analysis.
To analyze filament length, density and orientation, density of elastic energy and compressive strain as a function of the radial distance, we created radial bins spaced by 25 nm. In each bin, we collected the values of the various observables, as illustrated in Supplementary Figs. 1–3, 8 and 9. Local observables (density of elastic energy, filament density and orientation) have one value per segment. Their values were computed as the average of the observable in the bins. Non-local observables (filament length and compressive strain) have a single value per filament. Therefore, they were computed as the average of the observable for all filaments in the bin, weighted by the number of points of the filament in each bin.

### Estimation of the core radius.
All the parameters except the compressive strain were fitted as a function of the radial distance $r$ from the core center using the

following equation:

$$m(r) = \frac{1}{2}\left[(a+b) + (b-a)\tanh\left(\frac{r-r_0}{r_s}\right)\right] \quad (4)$$

$m(r)$ is either a decreasing function (for the filament density, filament orientation and the density of elastic energy) or an increasing function (for the filament length) with two saturation values set by the parameters $a$ and $b$ (Figs. 1e–g and 3f). The parameter $r_0$ corresponds to the radial distance for which the slope of the tangent line is maximum; the parameter $r_s$ defines the transition range between the two saturating values (Supplementary Fig. 5a). We used the parameter $r_0$ from the fit of the orientation as a measure of the core radius (Supplementary Fig. 5e, f).

**Polymerization force**. To estimate the polymerization force generated at the core membrane, the protrusive filaments, i.e., those in the immediate vicinity of the plasma membrane, were considered. Specifically, all the filament portions up to 10 nm away from the membrane were taken into account (Fig. 2a, b). A single filament growing perpendicularly to the membrane generates a force, $f_a$, given by[4]:

$$f_a = \frac{k_B T}{\delta} ln\left(\frac{C}{C_c}\right) \quad (5)$$

where $k_B T = 4.11 \times 10^{-21}$ J, $\delta = 2.75$ nm is the displacement induced by the addition of one actin monomer, $C$ is the concentration of actin monomers in solution and $C_c$ is the critical concentration. Using the values $C = 150\,\mu M$ as estimated in ref. [58] and $C_c = 0.06\,\mu M$ for the critical concentration at the plus end as measured in vitro[59,60], we found an upper limit of 11.4 pN for the stall force, while a typical value $C = 50\,\mu M$ would yield a force of 9.8 pN. Thus, the factor $ln(C/C_c)$ was set to 7, leading to a stall force of 10 pN for a single filament growing perpendicularly to the membrane. This value was increased by a factor $1/sin\,(\theta)$ for a filament having an orientation $\theta$ relative to the membrane[18] (Fig. 2d). Therefore, the total polymerization force, $F_{polym}^{core}$, was computed as follows:

$$F_{polym}^{core} = f_a \sum_{i=1}^{n_p} \frac{1}{\sin(\theta_i)} \quad (6)$$

where $n_p$ is the number of protrusive filaments and $\theta_i$ is the average orientation for the protrusive filament segment $i$ (Fig. 2e).

**Correlation length**. We computed the correlation of the orientation $C_s$ as a function of the distance $ds$ between two points on the same filament:

$$C(ds) = <t(s).t(s+ds)> \quad (7)$$

with $t(s)$ the tangent vector at arc-length $s$, and $<\,>$ the average over all filaments and all arc-lengths $s$. The number of points available to compute $C(ds)$ decreases rapidly upon increasing $ds$. Therefore, to keep statistical power, we kept only values of $ds$ associated with more than half of the maximal number of points (corresponding to $ds = 0$). To get the correlation length, $lc$, we then fitted $C(ds)$ to a decreasing exponential $\exp(-l/lc)$.

**Local curvature**. The local curvature was evaluated between two consecutive points using their respective tangent vector $t$ as follows:

$$\gamma_{i,i+1} = \frac{1}{a}\arccos(t_i.t_{i+1}) \quad (8)$$

where $a$ is their relative distance set to 3 nm in our segmentation procedure.

**Local elastic energy**. The local energy is proportional to the square of the local curvature and can be discretized as follows:

$$u_{i,i+1}^{elastic} = \frac{\kappa}{2}(\gamma_{i,i+1})^2 \quad (9)$$

where $\kappa$ is the bending modulus[30].

**Total elastic energy**. The total elastic energy inside the core was computed by summing the local energy over all the filament points that are within the radial distance domain $r \in [0, r_{core}]$, where $r_{core}$ is the core radius (Supplementary Fig. 8).

**Elastic force**. The elastic force pushing perpendicularly to the membrane is the derivative of the core elastic energy with respect to the height $h_{core}$ of the core:

$$F_{elastic}^{core} = \frac{dU_{elastic}^{core}}{dh_{core}} \approx \frac{\delta U_{elastic}^{core}}{\delta h_{core}} \quad (10)$$

with $\delta$ indicating small changes. Indeed, in the core, the average compressive strain of the filaments is $\epsilon_{core} \sim 0.016$ and therefore considered to be small. Assuming all elastic energy to be released in the resting state, we have $\delta U_{elastic}^{core} = U_{elastic}^{core}$ and $\delta h_{core} = \epsilon_{core} \times h_{core}$. Therefore, we find:

$$F_{elastic}^{core} \approx \frac{U_{elastic}^{core}}{\epsilon_{core} \times h_{core}} \quad (2)$$

Note that this is an order of magnitude estimate: since the actin network is assembled under pressure, it is likely that its resting state has a non-zero elastic energy, in which case the force is overestimated. The precise architecture of the network could also yield a prefactor in the relation between $\delta h_{core}$ and $\epsilon_{core}$. Lastly, the elastic energy could be underestimated: part of it might be stored in crosslinker elasticity and in tension of the non-bent segments of actin filaments.

**Pressure induced in the podosome core**. Assuming a circular shape of radius $r_{core}$ for the core, the pressure was computed as:

$$P = \frac{F_{elastic}^{core}}{\pi r_{core}^2} \quad (11)$$

In the rest of this section, the values used to compute the pressure from the cryo-ET data and the PFM force measurements reported in ref. [17] are detailed.

*From the tomograms*. The pressure induced by the compression of the actin filaments (labeled as "Elastic", Fig. 4a) and by the polymerization at the membrane (labeled as "Polymerization") were evaluated using the values for $r_{core}$ estimated from the fits of the orientation data (Fig. 1f, inset).

*From PFM force measurements*. The pressure $P_{PFM}$ was computed using the force value $F = 10.4 \pm 3.8$ nN reported in ref. [17] and $r_{core} = 203 \pm 38$ nm estimated from the average of 10 podosomes (black dashed line in Fig. 1f, inset). The error bar for this value was computed using the propagation of uncertainty method as follows:

$$\Delta P_{PFM} = \frac{2F_{elastic}^{core}}{\pi r_{core}^3}\Delta r_{core} + \frac{1}{\pi r_{core}^2}\Delta F_{elastic}^{core} \quad (12)$$

We found $P_{PFM} = 80.0 \pm 59.0$ kPa (Fig. 4b).

**Reporting summary**. Further information on research design is available in the Nature Research Reporting Summary linked to this article.

## Data availability

Data supporting the finding of this manuscript are available from the corresponding authors upon reasonable request. Five representative tomograms have been deposited in the EMDB under accession codes EMD-13666, EMD-13669, EMD-13671, EMD-13673, and EMD-13798. Source data are provided with this paper.

## Code availability

The analysis code, Podosome analyzer, is available on gitlab: https://gitlab.com/SergeDmi/podosome-demo[57].

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

## Acknowledgements

This work benefited from the assistance of Vanessa Soldan from the Multiscale Electron Imaging platform (METi) of the Centre de Biologie Intégrative (CBI). The authors thank Martin Lenz and Nicolas Minc for helpful discussions, W. Wan, P.S. Erdmann, S. Khavnekar, and V. Lucic for technical and computational assistance, and Daniel Baum for his help with the actin segmentation package. This work was supported by the Human Frontier Science Program (RGP0035/2016, I.M.P. and W.B.), la Fondation pour la Recherche Médicale (FRM DEQ2016 0334894, IMP), a CNRS Momentum fellowship (SD), l'Agence Nationale de la Recherche and Deutsche Forschungsgemeinschaft (ANR DFG 2020 JA-3038/2-1, M.J. and R.P.) and with financial support from ITMO Cancer of Aviesan on funds managed by Inserm (R.P.).

## Author contributions

M.J., R.P. designed the project. A.B., R.P. prepared the cells. S.B. vitrified the cells. M.J. performed cryo-FIB milling, cryo-ET, tomogram reconstruction, and segmentation, with help from J.S. for the cytochalasin D data. T.M. performed RIM imaging. M.J., J.H., A.P., R.P. designed and performed the experimental analysis. R.V., S.D. designed the theoretical analysis. J.H., S.D. performed the theoretical analysis. M.J., S.D., R.P. supervised the project. M.J., I.M.P., W.B., S.D., R.P. obtained funding. M.J., J.H., S.D., R.P. wrote the manuscript with input from the other authors.

## Funding

## Competing interests

The authors declare no competing interests.
