## [Peer Review File · Nature Communications]

Elasticity of podosome actin networks produces nanonewton protrusive forcesREVIEWER COMMENTS

Reviewer #1 (Remarks to the Author):

In this manuscript, Jasnin et al combine cryo-electron tomography, image analysis and theory to estimate the forces produced or supported by actin filaments in podosomes.

The results are noteworthy because it remains unclear how an actin network can generate net forces in the nN range, like in podosomes. Since individual filaments can produce only 1-10 pN, it would require hundreds of thousands filaments in podosomes to produce nN forces only by actin filament polymerization.

This work is significant to the field because, to our knowledge, it is the first quantitative analysis of the actin filaments present in podosomes. The lack of proper imaging data made it difficult for the field to validate or invalidate the different models for force production in podosomes. Thanks to their careful imaging and quantitative analysis, the authors were able to examine three hypothetical mechanisms that have been proposed for robust force generation by the actin network. They showed that two of these models (i.e. hundreds of thousands of filaments are present in podosomes, or filaments are almost parallel to the membrane) are not supported by the data. The data are compatible only with a model where elastic energy is stored in the bent actin filaments in podosomes.

While this conclusion is well supported by the data, the authors were not able to determine how the actin filaments get bent in the first place. In addition, the estimated stored elastic energy and forces are underestimates because the study accounts only for actin filament bending, and do not account for other elastic energies and forces that could be stored in other molecules (like proteins that connect filaments with each other or with the plasma membrane). That said, these points are clearly addressed in the discussion section of the manuscript, and answering these points are beyond the scope of this study.

We recommend the publication of this manuscript with minor revisions. Below are suggestions that could improve the manuscript:

1. The radial distance analysis (Fig S. 1) seems to depend on the orientation of the filaments and length. Could the method be updated to avoid any correlation between the average length and the orientation?
2. The method for radial distance analysis could be explained in more details in the method section.
3. L. 120-140: it would be interesting to compare the energies and forces calculated from the filaments' profiles obtained by cryo-ET with the energies and forces that would be expected from filaments only under thermal motion.
4. L. 204: the authors could discuss a third possibility mentioned earlier about the extra energy that could be stored in actin filament crosslinkers (or other proteins)
5. L. 43: "similar forces" could be changed in "forces in a similar range"
6. L. 44: the Dmitrieff et al and Ma et al papers could be cited here too.

7. L. 53-54: “low” and “large force regimes” could be explained for clarity
8. L. 105: “to evaluate the forces generated by actin ...”: it would be more precise to say “an upper limit of the forces ...”
9. L.125: clarify what the “average compressive strain” is for the non-physicists
10. L. 305: could point out that a cytoplasmic concentration of 150 uM is very large, and larger than more commonly accepted concentrations in the 50 uM range (which would reinforce the point the authors are making)
11. Add scale bars to Fig 2A, Fig 3A, and Fig S6B and C
12. Add units to Fig S4A and B

Reviewer #2 (Remarks to the Author):

In this paper Jasnin et al. employed cryo-ET to investigate the 3D architecture of human macrophage podosomes. There are a number of interesting findings. First, actin core has a 2-3 higher density of filaments with a more vertical orientation compared to filaments outside the core. Second, core filaments are shorter and more bent than the surrounding radial filaments, indicating that these bend filaments were under force. Additional back of the envelop calculations were conducted to estimate the contribution of polymerization force and elastic energy of the network. Generally, the experiments describing the actin assembly are convincing. The cryo-ET technique is very impressive. But my major concern is on the conclusion of paper which is a negative statement “the protrusive forces generated by podosomes cannot be explained by the sum of the actin polymerization forces at the core membrane...”. To prove the negative is much more difficult than demonstrating the positive. The estimated 615 ± 396 pN per podosome is relatively close to nN polymerization forces and we do not know whether such differences are still within the experimental errors or heterogeneities among different podosome sizes or stages. That been said, I think the conclusion is likely true and it is a logical one. I do not raise these concerns to request additional experiments as the authors could decide whether additional data collection is feasible. But the conclusion needs to be better justified. A number of key factors were not considered or sufficiently discussed that make the conclusion less rigorous than it could be at this stage.

1. For previous force measurement, cells were plated on elastic substrates (it is not clear whether that is the same or different from current preparation). Podosomes can push the substrate and form membrane bulges. Indeed in Fig 2 of ref 16, it was shown that force is proportional to height of membrane deformation, with 1 nm deformation on 40 nm Formvar membrane correspond to 10 nN but 0.2 nm deformation on 40 nm Formvar membrane correspond to 1 nN. It is not clear what is the extent of membrane deformation for the cryo-ET data and whether it necessarily correspond to maximal force as previously measured.

2. For estimation of maximal polymerization force, how does the variability of filament structures (branched or linear filaments or cross-linking) affect it?
3. What is the effect of neighbor podosome density and distances?
4. Dynamics of Podosomes. At different stages podosomes could generate higher or lower forces.

There are a few other questions that need to be clarified before it can be accepted:

1. How does the height of podosome in the tomogram compared to known podosome height (for instance obtained from 3D images of confocal microscopy)? Given the x-y dimension of the podosome (not diffraction-limited), I wonder whether the height of podosome is diffraction limited or not.
2. Line 151 "This can be explained by the disparity in the core size in our tomograms." not clear what it means.
3. Live cell imaging should be performed to show that 30 sec water treatment does not affect podosome structure significantly in order to validate the unroofing experiments. I suspect osmotic shock even for 30 sec may be sufficient to change the structure of podosome dramatically by disassembling some of the more dynamic populations of actin.

Reviewer #3 (Remarks to the Author):

In the paper "Elasticity of dense actin networks produces nanonewton protrusive forces" by Jasnin et al, the authors use cryo-electron tomography images in order to evaluate force generation from the actin network in podosomes. This evaluation is based on the 3D spatial organization of individual filaments, which is used as input of a theoretical model to make predictions about force generation. The model is based on linear elasticity. This approach is interesting and nicely shows how high resolution images can be used to generate quantitative estimations regarding mechanics of cells, however I see two main major points of concern:

1) the fact that the order of magnitude of the predicted force agrees with previous experimental measurements does not fully validate this approach. more detailed comparisons with experimental outputs, including perturbed conditions, is needed to support the results of the model and the validity of the approach. As an example, consider that here the force estimate scales with filaments density. It would be interesting to see what is the force if you decrease actin density experimentally, make an image of this system with cryo-electron tomography, estimate the force from this image (with the model) and then measure (or at least estimate) the same force experimentally. Alternatively, an idea would be to perturb filaments length and evaluate both experimentally and computationally how the force varies. Such a validation would strengthen the conclusions you make

2) this model supports the view that crosslinkers and molecular motors have a negligible contribution in force generation from actin filaments in podosomes. Is there any previous evidence in support of this view ? I believe that this would be a pretty strong point of the conclusions, therefore more discussion is needed to support this view.

Minor points

1. Can you use the model to assess the contribution of the horizontal filaments in the generation of the traction force that counterbalance the protrusion force? (lines 99-102)
2. Reference the hypotheses that: (i) hundreds of thousands of filaments push on the membrane; (ii) pushing angles are shallow. If these are not hypothesis coming from somewhere, then justify the rationale that lead to them, cause it is unclear in the current manuscript.
3. Motivate your model assumption that the podosome core behaves as a homogeneous elastic material
4. Is there any other biological system that relies on actin filaments architecture to generate force? If so, please compare and contrast in the discussion.

Reviewer #4 (Remarks to the Author):

In this work, the authors describe the in-situ ultrastructure of podosomes which are important force generating actin-based assemblies. Podosomes have garnered considerable attention particularly in immune cell migration. Though there is already a substantial understanding on the cell biological aspects of podosomes, a clear structural view of how actin filaments are arranged in podosomes is missing. This information could explain how forces are generated to protrude podosomes outwards.

In their manuscript the authors now describe the nanoscale architecture of podosomes in macrophages using cryo-electron tomography of lamellae obtained by focused ion beam milling. As expected from the authors, the generated cryo-ET data is of exceptional quality and provides fascinating insights into podosomal architecture. Specifically, their data unveils the 3-D architecture of actin filaments within podosomes, composing of core filaments surrounded by radial filaments. The authors quantitate the actin filament networks with respect to filament density, length and orientation. In contrast to what was expected by the authors, only a small number of actin filaments is oriented in a way to generate the protrusive force required to push against the membrane. However, the authors observe bent actin filaments in the podosome core, which could store elastic energy. They therefore propose a theoretical model to explain how the arrangement and bending of only a few filaments is able to produce the nanonewton forces, which were measured experimentally in earlier studies.

Overall, this is an interesting study with relevant findings, concerning the architecture of podosomal actin networks. However, while the experimental data is of great quality, the theoretical model is less exhaustive and appears oversimplified in the explanation of force generation. Specifically, the theoretical model is entirely based on the observation of bent filaments, however neglects other aspects of the network that one would expect to contribute to force generation in podosomes (or when establishing the required counterforce to avoid actin filaments simply pushing inwards into the cell). For example, it does not become clear from the paper how bent filaments in the core of the podosomes are formed and stabilized? Podosomes are N-WASP dependent protrusions, meaning one would expect a dendritic actin network formed by the Arp2/3 complex. Additionally, filaments are potentially cross-linked or maybe bundled within the podosomes.

If the forces and energy calculated in the presented theoretical model match the experimentally reported values without taking into account any cross linkers or other actin binding proteins, does this mean their contribution is negligible? The authors need to elaborate on these aspects, also with respect to the molecular differences of perpendicular (in the core) and radial filaments? Here the authors should discuss what other proteins might stabilize these assemblies and how their incorporation in the model might influence the obtained forces. For example, the authors have already analyzed the branch junction distribution in dense actin networks in their previous papers. I would encourage them to extend their analysis of the podosomes with this aspect.

Additional points:

1. The paper describes force generation in podosomal actin networks, and while it is correct that similar mechanisms can be expected in other actin networks, the manuscript does not unambiguously show this. Hence, the title should be more specific in stating that the analysis is focused on podosomal actin filament structures.
2. The observation of bent filaments is key to the derived theoretical model. Figure 3B currently shows examples of bent filaments both outside as well inside the core. What is missing, is a correlation of filament orientation with respect to the basal membrane, i.e. it does not become evident at the moment if all the bent filaments in the core are actually oriented in such an angle towards the membrane that would allow them to effectively contribute to force generation. This information needs to be included.

3. The authors reason that the force generated by the core filaments is counter-balanced by the radial filaments and maybe to a certain extent by the actin network present on top of the podosomes (which they can observe in the unroofed samples). Here the authors need to provide a bit more detail on how this could work specifically (for example also again considering the presence of crosslinking proteins):

a. Would it be possible to quantify the counterbalancing force produced by radial filaments? By knowing these values, one can easily tell if only radial filaments are sufficient to counter balance the forces generated by core of the podosomes.

b. The diameter of podosomes is variable. Does the number of radial filaments increase with increase in the diameter of the core?

Methodological comments:

4. There are no fluorescence images presented in this work, making it difficult to understand how exactly podosomes were targeted for ion-beam milling. Have all podosomes been imaged over holes in the carbon film or are they also located on continuous carbon (which could influence the SNR and hence accuracy of filament tracking)?

5. In line with the comment above, fluorescent images of macrophages should be added to provide readers with a better understanding of podosome abundance, distribution and size.

6. As stated in the methods, actin filaments of length shorter than 50-60nm are removed.

I understand that the removal of short filaments is probably necessary in order to remove false positive filament traces. However, do the authors believe that such short filaments do not provide any force to podosome formation or are otherwise important? A short consideration on this should be provided in the manuscript

7. In this context, it will be interesting to know how accurate the filament tracking works and if there is any confidence metric available for individually tracked filaments. This seems to be particularly important for filaments being oriented orthogonally to the tilt axis. Can the authors provide any indication on how many filaments their tracking approach might miss?

8. The tracking seems to be otherwise very clean. Out of curiosity, are there any additional cleaning (i.e. manual removal) steps involved in the filament tracking beyond removing short filaments?

9. All tomograms analysed in this manuscript should be deposited in the EMDB (e.g. as binned data) so that interested readers can get a better appreciation of the data.

We thank the reviewers for their valuable comments, and the editor for giving us the opportunity to revise the manuscript. We addressed all the reviewers' comments in the detailed point-by-point response provided below.

In short, we emphasized in the manuscript that crosslinkers are also expected to contribute to the storage of elastic energy in the podosome core. To illustrate filament deformation more comprehensively, we introduced a new parameter, the orientational correlation length. We provided a range for the polymerization forces generated at the membrane, using both the lower and upper limits of the polymerization force of an actin filament. We performed the same analysis on unroofed podosomes, which gave similar results as for the *in situ* data, indicating that the unroofing procedure does not significantly affect the architecture of the network. To address experimentally whether a decrease in actin density can be correlated with a decrease in protrusion force generation, we analysed podosomes from unroofed cells treated with cytochalasin D, which we have previously shown to abrogate force generation (Labernadie et al. Nat Commun 2014). The remaining compressive strain of the podosome core in cytochalasin D treated podosomes was about half of that of control podosomes, which is therefore correlated to the loss of protrusion force. In addition, we corrected the filament length that had been slightly overestimated in our analysis, which resulted in small modifications of the compressive strains, elastic forces (~30 nN on average), pressures (~200 kPa) and radial tensions (~50 mN/m).

We prepared a revised version of the manuscript with changes highlighted in yellow, which follows the guidelines of Nature Communications. Two new contributing authors were added for their help with tilt-series collection and reconstruction (Jonathan Schneider) and super-resolved macrophage imaging (Thomas Mangeat), following reviewers' comments.

Reviewer #1 (Remarks to the Author):

In this manuscript, Jasnin et al combine cryo-electron tomography, image analysis and theory to estimate the forces produced or supported by actin filaments in podosomes.

The results are noteworthy because it remains unclear how an actin network can generate net forces in the nN range, like in podosomes. Since individual filaments can produce only 1-10 pN, it would require hundreds of thousands filaments in podosomes to produce nN forces only by actin filament polymerization.

This work is significant to the field because, to our knowledge, it is the first quantitative analysis of the actin filaments present in podosomes. The lack of proper imaging data made it difficult for the field to validate or invalidate the different models for force production in podosomes. Thanks to their careful imaging and quantitative analysis, the authors were able to examine three hypothetical mechanisms that have been proposed for robust force generation by the actin network. They showed that two of these models (i.e. hundreds of thousands of filaments are present in podosomes, or filaments are almost parallel to the membrane) are not supported by the data. The data are compatible only with a model where elastic energy is stored in the bent actin filaments in podosomes.

While this conclusion is well supported by the data, the authors were not able to determine how the actin filaments get bent in the first place. In addition, the estimated stored elastic energy and forces are underestimates because the study accounts only for actin filament bending, and do not account for other elastic energies and forces that could be stored in other molecules (like proteins that connect filaments with each other). That said, these points are clearly addressed in the discussion section of the manuscript, and answering these points are beyond the scope of this study.

We recommend the publication of this manuscript with minor revisions. Below are suggestions that could improve the manuscript:

We thank the reviewer for these encouraging comments and the valuable suggestions below.

1. The radial distance analysis (Fig S. 1) seems to depend on the orientation of the filaments and length. Could the method be updated to avoid any correlation between the average length and the orientation?

To compute the filament length as a function of the radial distance, we need to average the length of all the filaments inside a bin. To do this, it is necessary to give each filament a weight proportional to the number of points it has inside that bin; otherwise, a filament with a single segment in a bin would have the same weight as a filament entirely in the bin. As a result, the weighting of a filament depends indirectly on its orientation, whereas its measured length does not. We modified the legend of Supplementary Fig. 1 and the methods section to clarify this point.

2. The method for radial distance analysis could be explained in more details in the method section.

We agree with the referee that the method description for the radial analysis lacked detail, and therefore expanded this section in the methods.

3. L. 120-140: it would be interesting to compare the energies and forces calculated from the filaments' profiles obtained by cryo-ET with the energies and forces that would be expected from filaments only under thermal motion.

We thank the reviewer for this valuable input. The energy of filaments under thermal motion should be $k_B T/2$ per degree of freedom; in other words, if we were to represent filament shape in Fourier space, each frequency mode would have an energy of $k_B T/2$. Even in theory, it is not clear at which frequency to cut off the summation of the Fourier modes. In practice, we are limited by the microscope resolution and the segmentation method. Therefore, it is not possible to determine the energies and forces that would be expected from filaments only under thermal motion. To allow a better understanding of the fact that filaments are not simply under the effect of thermal motion, we computed the orientational correlation length, which should be the persistence length of an actin filament if it were merely under thermal motion (Isambert et al. J Biol Chem 1995). We found that the correlation length is shorter in the core (1.68 μm) than outside (2.41 μm), and in both cases several times smaller than the persistence length of an actin filament (around 10 μm). Therefore, actin filaments are actively deformed in the podosome. We now use the correlation lengths instead of the degree of compression to describe filament deformation, as it is more easily understandable. These results are now discussed in the main text (p. 5, l. 138-141), and the correlation lengths shown in Fig. 3e.

4. L. 204: the authors could discuss a third possibility mentioned earlier about the extra energy that could be stored in actin filament crosslinkers (or other proteins).

Indeed, our results imply that the core filaments do not act independently but are mechanically linked to each other. The Arp2/3 complex, as well as fascin and L-plastin, are present in the core (Van Audenhove et al. *Biochim Biophys Acta* 2015; De Clercq et al. *PLoS One* 2013) and are likely to participate in network connectivity. α -actinin and filamin-A presumably connect the radial filaments to each other, but also possibly to the core filaments (van den Dries et al. *Nat Commun* 2019; Guet et al. *J Biol Chem* 2012). Finally, actin filaments have also been proposed to be connected to the substrate at the adhesion ring via integrins, talin and vinculin (Bouissou et al. *ACS Nano* 2017). All of these actin interactors therefore represent mechanical connections that are also subjected to forces and can also store elastic energy. Following the reviewer's suggestion, we added this discussion on the potential role of other proteins, especially crosslinkers, in the manuscript (p. 8, l. 239-245). It is important to note that integrating these proteins in our model would require knowing the number of crosslinkers present in podosomes and being able to evaluate how much energy is stored in each of them, which is currently unknown.

5. L. 43: "similar forces" could be changed in "forces in a similar range"

We reformulated accordingly.

6. L. 44: the Dmitrieff et al and Ma et al papers could be cited here too.

This was done.

7. L. 53-54: "low" and "large force regimes" could be explained for clarity

We now clarify these regimes.

8. L. 105: "to evaluate the forces generated by actin ...": it would be more precise to say "an upper limit of the forces ..."

We now mention in this paragraph that 10 pN is an upper limit.

9. L.125: clarify what the "average compressive strain" is for the non-physicists

The compressive strain, a dimensionless number between 0 (undeformed network) and 1 (fully compressed network), is computed as 1 minus the ratio between the end-to-end distance, c , and the filament length, L , and is now introduced later for the evaluation of the elastic force. We provided more information to clarify this term (p. 6, l. 160-165 and Supplementary Fig. 9).

10. L. 305: could point out that a cytoplasmic concentration of 150 μ M is very large, and larger than more commonly accepted concentrations in the 50 μ M range (which would reinforce the point the authors are making)

We thank the reviewer for this comment. The upper limit for the polymerization force of an actin filament is now estimated using both concentrations, and remains very close to 10 pN (p.13-14, l. 403-404). In addition, we now estimate the force generated by actin polymerization at the core membrane using both the lower (1 pN) and upper (10 pN) limits of the polymerization force (p. 5,

I. 119-124, Figs. 2d and 4a). These values are also overestimated since we consider that all the filaments grow concomitantly. We note that the value obtained with the upper limit of 10 pN was previously computed using the average orientation of the entire filament instead of the orientation of the filament segments close to the membrane. This was corrected, resulting in a mean value of 564 pN instead of 615 pN.

11. Add scale bars to Fig 2A, Fig 3A, and Fig S6B and C

A scale bar was added to Fig. 2b and Supplementary Fig. 6b. Figs. 2a, 3a and Supplementary Fig. 6c were rendered in a 3D perspective, which prevents the use of a scale bar. A scale reference was added in the legend of these figures.

12. Add units to Fig S4A and B

This was done.

Reviewer #2 (Remarks to the Author):

In this paper Jasnin et al. employed cryo-ET to investigate the 3D architecture of human macrophage podosomes. There are a number of interesting findings. First, actin core has a 2-3 higher density of filaments with a more vertical orientation compared to filaments outside the core. Second, core filaments are shorter and more bent than the surrounding radial filaments, indicating that these bend filaments were under force. Additional back of the envelop calculations were conducted to estimate the contribution of polymerization force and elastic energy of the network. Generally, the experiments describing the actin assembly are convincing. The cryo-ET technique is very impressive. But my major concern is on the conclusion of paper which is a negative statement “the protrusive forces generated by podosomes cannot be explained by the sum of the actin polymerization forces at the core membrane...”. To prove the negative is much more difficult than demonstrating the positive. The estimated 615 ± 396 pN per podosome is relatively close to nN polymerization forces and we do not know whether such differences are still within the experimental errors or heterogeneities among different podosome sizes or stages.

Following the reviewer's comment and comment #10 from Reviewer #1, we would like to clarify this point. What we called "actin polymerization force" in the first version of the manuscript is an upper bound, probably largely overestimated. Indeed, to evaluate this value we considered that every filament in the vicinity of the membrane polymerizes against it, and that each filament applies a force of 10 pN, and not 1 pN, a common lower bound, which would result in a difference of one to two orders of magnitude below the experimental values. We now estimate the polymerization forces under these two assumptions (p. 5, I. 119-124).

That been said, I think the conclusion is likely true and it is a logical one. I do not raise these concerns to request additional experiments as the authors could decide whether additional data collection is feasible. But the conclusion needs to be better justified. A number of key factors were not considered or sufficiently discussed that make the conclusion less rigorous than it could be at this stage.

1. For previous force measurement, cells were plated on elastic substrates (it is not clear whether that is the same or different from current preparation). Podosomes can push the substrate and form membrane bulges. Indeed in Fig 2 of ref 16, it was shown that force is proportional to height of membrane deformation, with 1 nm deformation on 40 nm Formvar membrane correspond to 10 nN but 0.2 nm deformation on 40 nm Formvar membrane correspond to 1 nN. It is not clear what is the extent of membrane deformation for the cryo-ET data and whether it necessarily corresponds to maximal force as previously measured.

We thank the reviewer for this valuable comment. The protrusion forces mentioned in this study were indeed evaluated on 30 nm-thick Formvar (Proag et al. ACS Nano 2015) and not on the 20 nm-thick carbon films that were used for cryo-electron tomography (cryo-ET). To evaluate the forces generated on carbon, we imaged by atomic force microscopy (AFM) a 20-nm carbon film onto which macrophages were let to adhere. Unfortunately, we could not identify any bulges reminiscent of podosome-induced deformations. This means that the podosomes do not produce forces high enough to cause deformations that can be measured by AFM. Of note, Formvar Young's modulus is about 2 GPa whereas Young's modulus values of carbon vary from <100 GPa to about 1000 GPa in the literature. For example, measurements of thin carbon membranes using AFM by Suk et al. (Carbon 2012) yielded values around 200 GPa. Given such a value, a 10 nN force on a 20 nm-thick carbon film would result in a 0.28 nm deformation, which is in the order of magnitude of the axial resolution of our AFM and thus cannot be measured accurately. To result in a 10-nm high deformation, equivalent to those observed on Formvar, a podosome should produce a 353 nN force, which is probably above the maximal force that can be produced by podosomes. To be able to answer this question, we will therefore need to develop a substrate compatible with both protrusion forces measurements and cryo-ET, but this would, in our opinion, merit a whole technological story in itself. To avoid any confusion, we now explicitly state in the manuscript whenever we mention the protrusion forces that they were measured on Formvar.

2. For estimation of maximal polymerization force, how does the variability of filament structures (branched or linear filaments or cross-linking) affect it?

Actin polymerization factors such as formins should not alter the reaction free energy of actin assembly, but rather the kinetics (Jegou, Carlier and Romet-Lemonne Nat Commun 2013). Therefore, whether actin forms linear or branched filaments should not directly affect the polymerization force. However, by nucleating filaments at specific angles with respect to the basal membrane, the mode of polymerization could have an influence. We already take this possibility into account by considering the angle of actin filaments with the membrane (Dmitrieff and Nédélec J Cell Biol 2016). Crosslinking slightly shifts the reaction thermodynamics towards a higher free energy of reaction (by adding the crosslinker binding energy), but this effect can be assumed to be small compared to the $\sim 10k_B T$ provided by ATP hydrolysis per actin monomer. In the main text, we clarified and discussed in more detail the implications of actin architecture on the Brownian ratchet model (p. 6, l. 173-175).

3. What is the effect of neighbor podosome density and distances?

It is already known, mainly from scanning electron microscopy images, that neighbouring podosomes are connected by actin filaments. Furthermore, we have previously shown that neighbouring podosomes, if their distance is less than 2 μm , are synchronous in their actin

dynamics and protrusion force. The organization of actin filaments between more or less closely spaced podosomes is therefore an intriguing question. Unfortunately, the magnification we used to reach an optimal resolution to analyse the organization of actin in podosomes provides tomograms with an edge length of 1.56 μm in x- and y-axis. This is suitable for capturing the architecture of single podosomes (as in the *in situ* data; see Fig. 1a for example), but more rarely of two adjacent podosomes as the average distance between podosomes is 1.8 μm (Proag et al. ACS Nano 2015). Our only tomogram of several actin cores is shown in Supplementary Fig. 6a. In this tomogram, the network between very close cores appears to be denser than between more distant podosomes, but statistical analysis to detect differences in architecture as a function of proximity to neighbours would require a much larger sample size.

4. Dynamics of Podosomes. At different stages podosomes could generate higher or lower forces. Indeed, podosomes are highly dynamic structures which grow and oscillate before disappearing after a few minutes, and it would be particularly informative to be able to discriminate between nascent and more mature podosomes. Here, we captured podosomes from different cells that are all large, and likely mature, podosomes. In order to only observe nascent podosomes, we should theoretically be able to synchronize podosome formation by Src kinase inhibitor washout (as in Cervero et al. Methods Mol Biol 2013) but to do this successfully for cryo-ET exploration is a technical challenge which would require long and uncertain development.

There are a few other questions that need to be clarified before it can be accepted:

1. How does the height of podosome in the tomogram compared to known podosome height (for instance obtained from 3D images of confocal microscopy)? Given the x-y dimension of the podosome (not diffraction-limited), I wonder whether the height of podosome is diffraction limited or not.

The podosome height evaluated by atomic force microscopy is 578 ± 209 nm (Labernadie et al. PNAS 2010). It is therefore below the axial resolution of a confocal microscope (app. 800 nm).

2. Line 151 “This can be explained by the disparity in the core size in our tomograms.” not clear what it means.

We rephrased this sentence as follows: “The force per unit area of the core exhibits much less variation with an average value of $P = 202.4 \pm 29.5$ kPa, which suggests that the force of a podosome is regulated primarily by its size.”

3. Live cell imaging should be performed to show that 30 sec water treatment does not affect podosome structure significantly in order to validate the unroofing experiments. I suspect osmotic shock even for 30 sec may be sufficient to change the structure of podosome dramatically by disassembling some of the more dynamic populations of actin.

We agree with the reviewer that unroofing could perturb the system. Therefore, we studied the architecture of native podosomes, and used the unroofed data only to visualize the upper part of podosomes, which is absent from the *in situ* data due to the milling process.

To further investigate the effect of the unroofing procedure on podosome architecture, we analysed the cryo-ET data from the unroofed podosomes and compared them to the *in situ* measurements, rather than using live cell imaging, which does not provide sufficient resolution to resolve single actin filaments. The distribution of filament length, orientation, density and correlation length for 5 unroofed podosomes is presented in Supplementary Figures 7 and 10. Filament density in the unroofed condition is similar to that of the *in situ* data (Fig. 1e and Supplementary Fig. 7a). After unroofing the average filament orientation is comparable inside the core ($41 \pm 21^\circ$ against $47 \pm 22^\circ$) and the same outside the core ($23 \pm 21^\circ$) (Fig. 1f and Supplementary Fig. 7b). Filament lengths are also similar (107 ± 50 nm vs 111 ± 46 nm in the core, and 182 ± 137 nm vs 166 ± 120 nm outside) (Fig. 1g and Supplementary Fig. 7c). However, the correlation lengths in the core ($2.30 \mu\text{m}$) and outside the core ($2.89 \mu\text{m}$) indicate that filaments are less compressed after unroofing than in their native context (1.68 vs $2.41 \mu\text{m}$) (Fig. 3e and Supplementary Fig. 10), indicating that the unroofing may have released some of the constraints on the filaments. Therefore, podosome architecture is well preserved after unroofing but filament bending is affected.

Reviewer #3 (Remarks to the Author):

In the paper "Elasticity of dense actin networks produces nanonewton protrusive forces" by Jasnin et al, the authors use cryo-electron tomography images in order to evaluate force generation from the actin network in podosomes. This evaluation is based on the 3D spatial organization of individual filaments, which is used as input of a theoretical model to make predictions about force generation. The model is based on linear elasticity. This approach is interesting and nicely shows how high resolution images can be used to generate quantitative estimations regarding mechanics of cells, however I see two main major points of concern:

1) the fact that the order of magnitude of the predicted force agrees with previous experimental measurements does not fully validate this approach. more detailed comparisons with experimental outputs, including perturbed conditions, is needed to support the results of the model and the validity of the approach. As an example, consider that here the force estimate scales with filaments density. It would be interesting to see what is the force if you decrease actin density experimentally, make an image of this system with cryo-electron tomography, estimate the force from this image (with the model) and then measure (or at least estimate) the same force experimentally. Alternatively, an idea would be to perturb filaments length and evaluate both experimentally and computationally how the force varies. Such a validation would strengthen the conclusions you make

We thank the reviewer for this suggestion. To decrease actin density experimentally, we treated cells with cytochalasin D, which we have previously shown to almost completely abolish protrusion force generation (Labernadie A. et al. Nat Commun 2014), unroofed and cryo-fixed the cells. Of note, as detailed in the answer to Reviewer #2's last question, podosome architecture is well preserved after the unroofing procedure (filament bending however is affected). Observations by fluorescence microscopy in Labernadie A. et al. (Nat Commun 2014) suggested that the network of radial filaments disappeared after cytochalasin D treatment. This was confirmed by our cryo-ET observations. Given that our current model postulates a force balance between core

protrusion and traction at the adhesion ring mediated by radial filaments, the disappearance of actin radial filaments in cells treated with cytochalasin D could explain the loss of protrusion force. Analysis of the architecture of the remaining cores further revealed that they are composed of less dense filament networks. More importantly, the compressive strain in these podosomes is decreased by 50%, compared to control unroofed cells *in situ*, which is therefore correlated to the loss of protrusion force. These results are now shown in Supplementary Figures 11 and 12 and discussed in the manuscript (p. 8, l. 215-231). A milder perturbation, allowing both force measurement and network analysis from *in situ* cryo-ET data, has yet to be identified but is a promising target for future work.

2) this model supports the view that crosslinkers and molecular motors have a negligible contribution in force generation from actin filaments in podosomes. Is there any previous evidence in support of this view ? I believe that this would be a pretty strong point of the conclusions, therefore more discussion is needed to support this view.

We thank the reviewer for this interesting point. In our model, considering only filament bending gives the correct order of magnitude for the forces, but crosslinkers are expected to contribute as well, as suggested by theoretical modelling of actin networks that generate forces during endocytosis in yeast (Ma and Berro Plos Comp Biol 2018). It should be noted that, while the contribution of crosslinking to force generation by podosomes is expected, it has not yet been investigated. We now discuss it in the results and discussion sections (p. 6, l. 154-156, p. 7, l. 186-192 and p. 8, l. 239-245).

Minor points

1. Can you use the model to assess the contribution of the horizontal filaments in the generation of the traction force that counterbalance the protrusion force? (lines 99-102)

In this manuscript, we define three actin networks in the podosome according to their locations and angles relative to the membrane: the core filaments, the radial filaments and the horizontal filaments. The horizontal filaments are present on top of the core (see Supplementary Fig. 6). From our observations, the horizontal filaments appear as a continuation of the radial filaments, and we believe that this set of radial + horizontal filaments is linked to the plasma membrane at the level of the adhesion ring (Bouissou et al. ACS Nano 2017). With less talin or vinculin, essential links between integrins and actin filaments, there is less traction and protrusion force at podosomes (Bouissou et al. ACS Nano 2017). Without the radial filaments (cytochalasin D condition), the podosome no longer generates forces (Labernadie et al. Nat Commun 2014 and Supplementary Figs. 11 and 12). We note that horizontal filaments, even under tension, would have a zero projected force on the vertical axis. Therefore, they should not directly contribute to the force balance.

2. Reference the hypotheses that: (i) hundreds of thousands of filaments push on the membrane; (ii) pushing angles are shallow. If these are not hypothesis coming from somewhere, then justify the rationale that lead to them, cause it is unclear in the current manuscript.

The widely accepted models of force generation in the literature indicate that a single actin filament generates pN force that depends on the concentration of free actin monomers and the angle between the filament and the plasma membrane (Dmitrieff and Nedelec J Cell Biol 2016).

Based on this model, we hypothesized that either hundreds to thousands of filaments are applying polymerization forces on the membrane or that fewer filaments are pushing, but with shallow angles. We rewrote this paragraph to clarify these hypotheses (p. 4, l. 111-114).

3. Motivate your model assumption that the podosome core behaves as a homogeneous elastic material

In the revised manuscript, we now mention that we follow the dominant assumption in the literature (Broedersz and MacKintosh Rev Mod Phys 2014). In addition, the overall small compression of the core is consistent with the elastic material being linear (p. 6, l. 166-168).

4. Is there any other biological system that relies on actin filaments architecture to generate force? If so, please compare and contrast in the discussion.

Of course, the different types of actin filament organization most likely have implications for their ability to generate greater or lesser forces. The novelty of our work is to highlight the curvature of the filaments, to measure it with high precision, and to propose an explanation of how the elastic energy stored in the network allows force generation. To our knowledge, only the recent article by Akamatsu et al. (Elife 2020) highlights the presence of curved filaments during mammalian endocytosis. It is important to note that the forces involved in this model are much smaller than those generated during endocytosis in yeast or in the podosome. Following the reviewer's suggestion, we added in the discussion that our work naturally allows us to assume that elastic energy storage in actin filament networks may play a role in force generation by many other structures, including yeast endocytosis or the lamellipodium.

Reviewer #4 (Remarks to the Author):

In this work, the authors describe the in-situ ultrastructure of podosomes which are important force generating actin-based assemblies. Podosomes have garnered considerable attention particularly in immune cell migration. Though there is already a substantial understanding on the cell biological aspects of podosomes, a clear structural view of how actin filaments are arranged in podosomes is missing. This information could explain how forces are generated to protrude podosomes outwards.

In their manuscript the authors now describe the nanoscale architecture of podosomes in macrophages using cryo-electron tomography of lamellae obtained by focused ion beam milling. As expected from the authors, the generated cryo-ET data is of exceptional quality and provides fascinating insights into podosomal architecture. Specifically, their data unveils the 3-D architecture of actin filaments within podosomes, composing of core filaments surrounded by radial filaments. The authors quantitate the actin filament networks with respect to filament density, length and orientation. In contrast to what was expected by the authors, only a small number of actin filaments is oriented in a way to generate the protrusive force required to push against the membrane. However, the authors observe bent actin filaments in the podosome core, which could store elastic energy. They therefore propose a theoretical model to explain how the arrangement and bending of only a few filaments is able to produce the nanonewton forces, which were measured experimentally in earlier studies. Overall, this is an interesting study with relevant

findings, concerning the architecture of podosomal actin networks. However, while the experimental data is of great quality, the theoretical model is less exhaustive and appears oversimplified in the explanation of force generation. Specifically, the theoretical model is entirely based on the observation of bent filaments, however neglects other aspects of the network that one would expect to contribute to force generation in podosomes (or when establishing the required counterforce to avoid actin filaments simply pushing inwards into the cell).

We thank the reviewer for these encouraging comments. Our straightforward model aims to propose an explanation for the magnitude of the protrusion forces generated by podosome cores. To this end, we show that the energy stored in filaments is sufficient to justify the order of magnitude of the podosome protrusion forces. Due to the lack of precise data on the amount of energy stored in proteins linking the actin filaments or connecting actin filaments to the plasma membrane, we did not include them in our order-of-magnitude calculations, which we now clarify in the discussion (p. 8, l. 239-245). Of course, many questions remain to be answered, such as the importance of actin crosslinkers and branches for force generation, or the regulation of polymerization and coupling of radial filaments, but we believe that these aspects are outside the scope of this manuscript and will be the subject of further studies.

For example, it does not become clear from the paper how bent filaments in the core of the podosomes are formed and stabilized?

This is indeed a fascinating question that our results bring to light. Until now, filament polymerization at the podosome core was thought to be mediated by WASP and Arp2/3 complex at the plasma membrane. Our results suggest that the few filaments in contact with the plasma membrane can no longer polymerize because they are subjected to forces above the stall force of polymerization. In the discussion section (p. 9, l. 251-264), we propose two new hypotheses to explain this phenomenon: (i) that filaments could grow in the absence of load, and then be put under load by myosins present in the radial actin cables, or (ii) that filaments could grow bent, protected by the existing network which would bear the load. In case (ii), an additional active (energy consuming) mechanism would likely be required to obtain the observed architecture. In both cases, the load-bearing filaments are out of equilibrium, since they bear a force larger than their stall force. Therefore, filaments have to be stabilized kinetically, e.g. by capping proteins, or by slow depolymerization kinetics relative to the timescale of force exertion, which is of the order of minutes (Labernadie et al. Nat Commun 2014).

Podosomes are N-WASP dependent protrusions, meaning one would expect a dendritic actin network formed by the Arp2/3 complex. Additionally, filaments are potentially cross-linked or maybe bundled within the podosomes. If the forces and energy calculated in the presented theoretical model match the experimentally reported values without taking into account any cross linkers or other actin binding proteins, does this mean their contribution is negligible? The authors need to elaborate on these aspects, also with respect to the molecular differences of perpendicular (in the core) and radial filaments? Here the authors should discuss what other proteins might stabilize these assemblies and how their incorporation in the model might influence the obtained forces.

We thank the reviewer for this valuable input, which converges with those of the other reviewers. The contribution of other actin binding proteins could also be significant for podosome force

generation: crosslinkers and Arp2/3 complexes could add extra degrees of freedom where elastic energy could be stored (see also response #2 to Reviewer #3). This is in agreement with our main message, i.e. that the properties of the entire network, rather than just its interface with the membrane, are important for force generation. We now emphasize in the manuscript that the force estimation gives an order of magnitude estimate for the protrusion forces generated by podosomes (p. 6-7, l. 173-192) and discuss the possibility that actin crosslinkers in the core and the radial filaments also store elastic energy (p. 8, l. 235-245).

For example, the authors have already analyzed the branch junction distribution in dense actin networks in their previous papers. I would encourage them to extend their analysis of the podosomes with this aspect.

Finding branch junctions in podosomes was indeed one of our goals. Unfortunately, the podosome cores are so dense that identifying branch junctions by template matching proved to be much more difficult than expected, and did not yield satisfactory results. The unroofed data, which has a better signal-to-noise ratio, was also analysed, without more success. The detection of branches and cross-linkers in podosomes will require the development of new detection and classification tools (e.g., deep learning algorithms), which is beyond the scope of this paper.

Additional points:

1. The paper describes force generation in podosomal actin networks, and while it is correct that similar mechanisms can be expected in other actin networks, the manuscript does not unambiguously show this. Hence, the title should be more specific in stating that the analysis is focused on podosomal actin filament structures.

We agree with the reviewer that the focus on podosomes should be highlighted. Accordingly, the title was changed to: "Elasticity of podosome actin networks produces nanonewton protrusive forces"

2. The observation of bent filaments is key to the derived theoretical model. Figure 3B currently shows examples of bent filaments both outside as well inside the core. What is missing, is a correlation of filament orientation with respect to the basal membrane, i.e. it does not become evident at the moment if all the bent filaments in the core are actually oriented in such an angle towards the membrane that would allow them to effectively contribute to force generation. This information needs to be included.

How the orientation of the filaments relative to the plasma membrane contributes to the protrusion force is an intriguing question. Actually, depending on the network architecture and especially on its density, one can imagine that both vertical and horizontal filaments can contribute to the protrusion force. To appreciate both the orientation and curvature of the filaments in the podosome, we show in Fig. 3a and Supplementary Movie 6 the actin filaments coloured as a function of their mean curvature. Plotting the curvature as a function of the filament orientation did not reveal any obvious correlation (Fig. 1 below), so it does not seem obvious to us to conclude on the importance of the angle. Of note, even if we were mistaken and should only take into account the vertical component of the compression, this would change our evaluation of the force by a factor of sine of the angle, i.e. a difference of only 30%.

Figure 1. Filament curvature as a function of its orientation relative to the plasma membrane.

3. The authors reason that the force generated by the core filaments is counter-balanced by the radial filaments and maybe to a certain extent by the actin network present on top of the podosomes (which they can observe in the unroofed samples). Here the authors need to provide a bit more detail on how this could work specifically (for example also again considering the presence of crosslinking proteins):

a. Would it be possible to quantify the counterbalancing force produced by radial filaments? By knowing these values, one can easily tell if only radial filaments are sufficient to counter balance the forces generated by core of the podosomes.

We were able to estimate the surface tension of the 2D meshwork of radial actin filaments that is required to balance the protrusion force (~ 50 mN/m) (p. 7-8, l. 205-214). However, while filaments buckle under compression, they barely stretch under tension (Lenz et al. Phys Rev Lett 2012). Therefore, there is no simple, direct estimate of their force from the cryo-ET data. Force balance implies that the tension of radial actin filaments is balanced by adhesion forces at the ring, exerted by adhesion complexes, in agreement with experimental data showing the importance of adhesion-associated proteins (talin, vinculin and paxillin) in force generation (Bouissou et al. ACS Nano 2017).

b. The diameter of podosomes is variable. Does the number of radial filaments increase with increase in the diameter of the core?

We performed a linear fit of the number of radial filaments as a function of the core radius and found the number to be $1729 (+/- 1143) + 1.6 (+/- 5.3) \times r_{\text{core}}$ with an r^2 of only 0.13, i.e this fit explains only 13% of the variance of the number of radial filaments. Therefore, there seems to be no correlation between the number of filaments outside the core and the size of the podosome.

Methodological comments:

4. There are no fluorescence images presented in this work, making it difficult to understand how exactly podosomes were targeted for ion-beam milling. Have all podosomes been imaged over holes in the carbon film or are they also located on continuous carbon (which could influence the SNR and hence accuracy of filament tracking)?

Two movies were added to show the distribution and dynamics of podosomes in human macrophages (Supplementary Movies 1 and 2). As podosomes cover the basal surface of the cell, we randomly targeted ventral parts of the cell using cryo-FIB milling. Cryo-ET data were collected on the carbon support film and not on top of the holes where no podosomes can be observed.

5. In line with the comment above, fluorescent images of macrophages should be added to provide readers with a better understanding of podosome abundance, distribution and size. Supplementary Movies 1 and 2 now illustrate that podosomes are distributed all over the basal surface of human macrophages.

6. As stated in the methods, actin filaments of length shorter than 50-60nm are removed. I understand that the removal of short filaments is probably necessary in order to remove false positive filament traces. However, do the authors believe that such short filaments do not provide any force to podosome formation or are otherwise important? A short consideration on this should be provided in the manuscript

In the podosome core, short filaments close to the plasma membrane would contribute to an increase in the polymerization force. However, for them to significantly increase the polymerization force (e.g., by a factor of 10), these filaments would have to be 10 times more numerous than those detected, which is not observed in the data (see Fig. 2 below for example).

Figure 2. Slice through podosome #1 in vicinity of the plasma membrane (left) and its superimposition with the filaments (right).

Within the full height of the podosome core, short filaments would increase actin filament density and connectivity, and thus, the estimated elastic force. However, since our method merely yields an order of magnitude estimate for the force, their contribution should not alter our results. We have added these points to the manuscript (p. 5, l. 127-129 and p. 7, l. 189-192).

7. In this context, it will be interesting to know how accurate the filament tracking works and if there is any confidence metric available for individually tracked filaments. This seems to be particularly important for filaments being oriented orthogonally to the tilt axis. Can the authors provide any indication on how many filaments their tracking approach might miss?

There is no confidence metric to assess the accuracy of the filament tracing in Amira. Because our cryo-ET data are intrinsically noisy, we unfortunately cannot reliably estimate how many filaments are missed. At the same time, we cannot properly assess the ground truth, and due to the complexity of our *in situ* data, it is not feasible to simulate them to provide a statistical metric of filament tracing accuracy.

The tomogram segmentation is performed iteratively, adjusting the extraction parameters until the filament traces match the tomographic data as closely as possible. Below certain correlation values, the procedure adds only tiny (false-positive) filaments which are then filtered by the length threshold. If the segmentation is not satisfactory to the expert eye and too many filaments are missed, the tomograms are discarded.

We acknowledge the limitation of Amira filament segmentation, especially with even higher-resolution data where actin monomers are resolved, and for which the cylinder template does not accurately represent the filament structure. Future solutions will consist in developing neural network algorithms to trace not only actin filaments but the entire network with its crosslinkers, bundlers and other short structural features.

8. The tracking seems to be otherwise very clean. Out of curiosity, are there any additional cleaning (i.e. manual removal) steps involved in the filament tracking beyond removing short filaments?

Yes, once the filament segmentation is satisfactory, careful manual inspection of the filaments is performed to remove false actin filaments (e.g., membrane, microtubules, intermediate filaments or noise arising from fiducials or dirt).

9. All tomograms analysed in this manuscript should be deposited in the EMDB (e.g. as binned data) so that interested readers can get a better appreciation of the data.

We deposited the 5 tomograms corresponding to the podosomes shown in the figures and movies in the EMDB under accession codes EMD-13666, EMD-13669, EMD-13671, EMD-13673 and EMD-13798.

REVIEWERS' COMMENTS

Reviewer #1 (Remarks to the Author):

The authors addressed all my comments.

Reviewer #2 (Remarks to the Author):

The authors have addressed all my questions and I support its publication.

Reviewer #3 (Remarks to the Author):

The authors fully addressed all my questions and improved the manuscript by adding new data, explaining their hypothesis and extending the discussion.

Reviewer #4 (Remarks to the Author):

The authors have addressed all comments and I can recommend the manuscript for publication in Nature Communications.